# Simulations of the Rotor-Stator-Cavity Flow in Liquid-Floating Rotor Micro Gyroscope

**DOI:** 10.3390/mi14040793

**Published:** 2023-03-31

**Authors:** Chunze Wang, Rui Feng, Yao Chu, Qing Tan, Chaoyang Xing, Fei Tang

**Affiliations:** 1China Unicom Digital Technology Company Limited, Beijing 100084, China; 2Research Institute for Frontier Science, Beihang University, Beijing 100191, China; 3Beihang Hangzhou Innovation Institute Yuhang, Hangzhou 310023, China; 4State Key Laboratory of Precision Measurement Technology and Instruments, Department of Precision Instrument, Tsinghua University, Beijing 100084, China; 5CISDI Group Co., Ltd., Chongqing 400013, China; 6Beijing Institute of Aerospace Control Devices, Beijing 100094, China

**Keywords:** microscale flow field, rotor-stator-cavity, Reynolds numbers, Reynolds stress model

## Abstract

When rotating at a high speed in a microscale flow field in confined spaces, rotors are subject to a complex flow due to the joint effect of the centrifugal force, hindering of the stationary cavity and the scale effect. In this paper, a rotor-stator-cavity (RSC) microscale flow field simulation model of liquid-floating rotor micro gyroscopes is built, which can be used to study the flow characteristics of fluids in confined spaces with different Reynolds numbers (*Re*) and gap-to-diameter ratios. The Reynolds stress model (RSM) is applied to solve the Reynolds averaged Navier–Stokes equation for the distribution laws of the mean flow, turbulence statistics and frictional resistance under different working conditions. The results show that as the *Re* increases, the rotational boundary layer gradually separates from the stationary boundary layer, and the local *Re* mainly affects the distribution of velocity at the stationary boundary, while the gap-to-diameter ratio mainly affects the distribution of velocity at the rotational boundary. The Reynolds stress is mainly distributed in boundary layers, and the Reynolds normal stress is slightly greater than the Reynolds shear stress. The turbulence is in the state of plane-strain limit. As the *Re* increases, the frictional resistance coefficient increases. When *Re* is within 10^4^, the frictional resistance coefficient increases as the gap-to-diameter ratio decreases, while the frictional resistance coefficient drops to the minimum when the *Re* exceeds 10^5^ and the gap-to-diameter ratio is 0.027. This study can enable a better understanding of the flow characteristics of microscale RSCs under different working conditions.

## 1. Introduction

Studies on the flow characteristics of rotor-stator-cavities (RSCs) play an important role in various industrial fields such as turbine machinery [1,2,3], disk drivers [4,5] and chemical stirrers [6,7,8]. Researchers have conducted extensive research on the enclosed viscous flow between rotational and stationary parallel disks by means of simulations, experiments and theoretical analyses. Daliy and Nece [9] took the lead in studying the flow of rotational disks and concluded four basic modes of rotor-stator flows, which are laminar flow with merged boundary layers (Regime I), laminar flow with unmerged boundary layers (Regime II), turbulent flow with merged boundary layers (Regime III) and turbulent flow with unmerged boundary layers (Regime IV). Gauthier et al. [10] studied the generation and spread of unstable disk flows in enclosed cavities through visualization methods and found that unstable fluctuations first appear in the stationary wall boundary layer as the Reynolds number (*Re*) increases. If *Re* continues to increase, such fluctuations gradually develop into annular waves and spiral waves that spread continuously from the edge to the center, and eventually develop into areas of turbulence. Schouvelier et al. [11,12] studied the correlation between *Re* and the gap-to-diameter ratio and the RSC flow field stability by photographing the wave structure and vortex structure in the flow field and mapped the flow fluctuations of disks in enclosed cavities. Watanabe et al. [13,14] carried out simulations and experiments to study the effects of radial slits and the size of such slits on the fluid structure in cavities. Serre et al. [15], through 3D direct numerical simulation, analyzed the relationship between the spatial structure and time of flows in annular and cylindrical cavities at different curvature and aspect ratios. Hara et al. [16] selected five disks of different radii to study the effects of the size of the radial gap and *Re* on the flow structure and drew the statistical map of morphological change nodes of the Taylor vortex and fluctuant vortex.

Nowadays, miniaturisation and portability of instruments and equipment have become a trend. The widespread use of MEMS technology has driven the market’s demand for micro gyroscopes [17,18,19], micro disks [5] and other micro rotating devices. As the devices get smaller, frictional resistance will cause not only higher energy consumption but also poorer performance and shorter service lives, which hinders the application of micro devices. Heating and increased power consumption caused by solid–liquid frictional resistance [20,21] are especially significant in liquid-floating micro gyroscopes. However, the changes of fluid flow, temperature distribution, torque and driving power between disks at micro scale have not been fully studied [11,12,13,14,22]. Therefore, it is necessary to investigate the flows of microscale rotational disks.

Three methods are mainly used to simulate turbulent flow, direct numerical simulation (DNS), large-eddy simulation (LES) and Reynolds-averaged Navier–Stokes (RANS). DNS and LES are regarded as the most accurate and trustworthy in unsteady high-Reynolds wall flows, but DNS is too expensive to be applied in complex project and LES just solve the turbulent scales of greater intensity. Due to its modest computing requirements, RANS has been the most popular method for industrial CFD applications over the past few decades, although errors introduced by modeling all turbulence scales would reduce predictive power. RANS decomposes the instantaneous motion that satisfies the dynamic equation into average motion and fluctuation motion, where the fluctuation item is represented by Reynolds stress. It solves only for the averaged quantities while the effect of all the scales of instantaneous turbulent motion is modelled by a turbulence model [23,24]. Among all the turbulence models, the RSM is the most consistent with the physical understanding of the RANS, considering the anisotropy of eddy viscosity. It is suitable for rotating flow, curved pipe flow, etc. [25].

In this paper, an RSC microscale flow field simulation model of liquid-floating rotor micro gyroscopes is established to calculate the flow field distribution at different *Re* numbers and gap-to-diameter ratios with the Reynolds stress model (RSM). The mean flow, turbulence statistics and frictional resistance coefficient in axial slits and the distribution law of the vortex structure in radial slits have been attained, which provides support for the design of liquid-floating rotor micro gyroscopes. Simultaneously, this work has great significance for the design and parameter setting of the devices with microscale RSC flow fields.

## 2. Model Parameters and Numerical Method

Figure 1 shows the section of the flow field in the *r*-*z* plane. The origin of the coordinate axis is at the center of the upper face of the disk. r represents radial distance from the origin. The radius and thickness of the disk are recorded as *R_d_* and *H_d_*, and those of the cavity as *R_c_* and *H_c_*, respectively. The size of the upper axial slit *H_u_* is the same as that of the lower axial slit *H_l_*. The disk rotates at the angular speed of *ω*, and the liquid in the cavity is water. At 20 °C, the kinetic viscosity coefficient *v* of water is 1.007 × 10^−6^ m^2^/s, and *Re*, local Reynolds number *Re_l_* and the gap-to-diameter ratio *G* are calculated in Equations (1)–(3).
(1)$$Re=\omega R_{d}^{2}/v,$$
(2)$${Re}_{l}=\omega r^{2}/v,$$
(3)$$G=H_{u}/R_{d}.$$

For the modelling in simulation, *H_u_* is set as 100 μm, 200 μm and 300 μm, and other parameters are as shown in Table 1.

The flow field model in this study is affected jointly by the rotational state of the rotor, the scale effect and the wall effect of the flow field, with complex flow characteristics and harsh requirements especially for the calculation of the turbulence. The RSM model can accurately simulate the flows in RSCs, especially at the points near the boundary of the disk where most of the turbulence is. It can describe in detail the near-wall turbulence without being affected by any hypothesis of eddy viscosity [26]. 

ANSYS Fluent was used for simulation. First, the 3D model was built. The radial scale of the simulated flow field is much larger than the axial scale, so the model was divided into upper and lower end caps, upper and lower ring domains, and cylindrical areas. Second, the grid independence verification had been done to ensure the efficiency and accuracy of the solution calculation. Each area was meshed separately with structured grid. The grids near the rotor wall and the cavity shell wall were refined. The grid orthogonality quality was used for evaluation. The grid quality distribution was concentrated at 0.8, indicating that the grid quality was better. Third, convenience conditions and initial conditions were set. The upper and lower end surfaces and side walls of the rotor were set as the rotation boundary and the rotation speed was set to 300 rpm, 1200 rpm, 3000 rpm, 6000 rpm, 12,000 rpm, etc. All the walls of the cavity shell were set as the static boundary. All walls were set as no-slip boundary conditions. The liquid in the model was set to be water at 20 °C. Fourth, the parameters of solver were set. Pressure, momentum, turbulent kinetic energy and turbulent diffusion rate adopted the standard, the second-order upwind, the first-order upwind and the first-order upwind, respectively. After discretization, the SIMPLE algorithm was used to solve the pressure coupling equation. The equations solved by Fluent are non-linear, and only by controlling the variable value in each iteration can the divergence caused by large difference be effectively avoided. It is regulated by the relaxation factor. After testing, when the relaxation factor of each calculation parameter was set to Table 2, the calculation was easy to converge. Last, iterative calculations were performed. The convergence standard of the residual value of each variable was set to 10^−6^. In addition, the velocity of a certain point in the model was selected as a reference. When it reached stability, the calculation result was considered to have converged.

In the subsequent study, all variables are normalised in the following way: $Z^{*}=z/H_{u}$, $r^{*}=r/R_{d}$, $V_{i}^{*}=V_{i}/\left(\omega \times r\right)$, $R_{ij}^{*}=v_{i}v_{j}/{\left(\omega \times r\right)}^{2}$, where *i* and *j* are any direction vector of the cylindrical-coordinate system *r*, *θ* and *z*, *Z** and *r** are the axial coordinate and circumferential coordinate after normalisation, $V_{\theta }^{*}$ and $V_{r}^{*}$ are the circumferential velocity and radial velocity after normalisation, *v_i_* and *v_j_* are the fluctuating velocity, $R_{ij}^{*}\left(i\neq j\right)$ is the Reynolds shear stress, $R_{ij}^{*}\left(i=j\right)$ is the Reynolds normal stress.

## 3. Results and Discussion

### 3.1. Mean Flow of the Axial Gap

Figure 2 illustrates the distribution of the circumferential velocity $V_{\theta }^{*}$ and the radial velocity $V_{r}^{*}$ along the z-axis when *Re* is 7.02 × 10^4^ and the gap-to-diameter ratio is 0.027. The axial velocity of the fluid at positions other than those around the rotor and near the edge of the lateral wall of the cavity is 0, which, therefore, is not given here. The radius *r** ranges from 0.067 to 0.967. *Z** is 0 at the rotational wall, and *Z** is 1 at the stationary wall. The velocity gradually decreases from the rotational boundary to the stationary boundary layer due to the non-slip boundary condition. The velocity gradient first decreases and then increases, and it changes more significantly at the position of a greater *r**. When *Z** < 0.1 (*z* < 20 μm), the circumferential velocity changes almost in the same way, and then the separation gradually occurs. According to the above results, there is no core region with a velocity gradient of 0, but the velocity gradient tends to 0 as the *Re_l_* increases. The centrifugal force generated by the rotating motion of the rotor causes the fluid near the rotational wall to be thrown out along the radius, so the radial velocity is a positive value. Following the law of conservation of mass, the fluid flows inward along the stationary wall, causing the radial velocity of the fluid near the stationary wall to have a negative value. As a result, the entire radial velocity is in an S-shaped distribution along the z-axis. Near the rotational wall, the position of the *V_r_** extremum remains almost unchanged, which is near *Z** = 0.1, while for the fluid near the stationary wall, the position of the *V_r_** extremum increases as *r** increases, which is offset to the stationary wall. According to the research by Daliy and Nece [9], the flow at this point is Regime II, which is the laminar flow with unmerged boundary layers. The boundary layer near the stationary wall is called the Ekman boundary layer [27], the thickness of which decreases as *r** increases, while the boundary layer near the rotational wall is called the Bödewadt boundary layer [27], the thickness of which remains almost unchanged.

Figure 3 displays the distribution curves of the circumferential velocity and radial velocity along the z-axis with the change of *Re* (1.75 × 10^3^ ≤ *Re* ≤ 7.02 × 10^4^). The axial velocity *V_z_** is still extremely low under this condition, so it is ignored. When *Re* = 7.02 × 10^3^ (at 1200 rpm rotational speed), the circumferential velocity is in a linear distribution along the z-axis. With the increase in *Re*, the flow of the fluid in the axial slit changes from the torsional Couette flow with a merged boundary layer to the laminar Batchelor flow with unmerged boundary layers, that is, it changes gradually from Regime I to Regime II [9]. When *Re* ≥ 1.75 × 10^4^ (at 3000 rpm rotational speed), the boundary layer gradually separates, and the separation becomes more significant as *Re* increases. The radial velocity *V_r_**, as a whole, retains its *S* shape, but its extremum increases as *Re* increases, and its position goes closer to the wall, with almost the same change law within the range of 0.4 ≤ *Z** ≤ 0.6.

Figure 4 shows the distribution curves of the circumferential velocity *V_θ_** and the radial velocity *V_r_** along the z-axis with the change of the gap-to-diameter ratio *G*. Under all three working conditions, the motion of the fluid is in Regime II, the laminar flow with unmerged boundary layers. The circumferential velocity *V_θ_**, when near the stationary wall (*Z** > 0.7), has almost the same change law, which is irrelevant to the gap-to-diameter ratio, but near the rotational wall, its gradient increases as the gap-to-diameter ratio increases. However, under all working conditions, the radial velocity *V_r_** is in an S-shaped distribution, and its extremum increases as the gap-to-diameter ratio increases, but its position along the axis is irrelevant to the gap-to-diameter ratio. Its extremum near the rotational wall is near *Z** = 0.12, and the same near the stationary wall is near *Z** = 0.85.

### 3.2. Turbulence Statistics in the Axial Gap

The distribution curves along the z-axis of the six components of the Reynolds stress tensor, *R_rr_**, *R_θθ_**, *R_zz_**, *R_rθ_**, *R_θz_** and *R_rz_**, at the given *Re* = 7.02 × 10^4^ and gap-to-diameter ratio *G* = 0.027 are shown in Figure 5. It can be seen from the figure that the Reynolds stresses of the turbulence are mainly concentrated near the boundary layer. The Reynolds stresses are all 0 in the central flow field area when *r** = 0.067, while almost all of them have a positive value at the other radii monitored. With the increase in *r**, the local Reynolds number *Re_l_* increases, the Reynolds stress slightly increases and the normal stress is higher than the shear stress.

Figure 6 shows the distribution curves of the six components of the Reynolds stress tensor along the z-axis at different *Re* when the gap-to-diameter ratio *G* = 0.027 and *r** = 0.8. It can be observed that the *Re* gradually decreases from the red line to the blue line, and the Reynolds stresses are of an approximately symmetric distribution with *Z** = 0.5 as the axis of symmetry. When *Re* ≤ 7.02 × 10^3^, the Reynolds normal stress decreases and then increases from the stationary boundary layer to the rotational boundary layer and has a lateral U-shaped distribution, while it is in an Σ-shaped distribution along the z-axis when *Re* ≥ 1.75 × 10^4^. The Reynolds shear stress changes similarly near the rotational boundary layer (0 ≤ *Z** < 0.1) and the stationary boundary layer (0.85 < *Z** ≤ 1) at different *Re* and increases as *Re* increases in the central area of the axial slit (0.1 ≤ *Z** ≤ 0.85), but is significantly lower than the normal stress.

Figure 7 illustrates the distribution curves of the six components of the Reynolds stress tensor along the z-axis at different gap-to-diameter ratios when *Re* = 7.02 × 10^4^ and *r** = 0.8. The normal stresses are in an Σ-shaped distribution and decrease as the gap-to-diameter ratio increases, but the three shear stresses have no such change. The Reynolds shear stress *R_rθ_** and *R_rz_** have negative values near the wall, and positive values near the central area of the slit, which is related to the centrifugal force and conservation of mass.

### 3.3. Anisotropy Analysis of Turbulence in the Axial Gap

The anisotropy invariant map (AIM) is an important method proposed by Lumley [28] to describe the morphology and structure of turbulence. Figure 8 shows a triangle bounded by three curves and their corresponding equations. The three purple curves indicate the limiting boundaries of the morphology of turbulent fluctuations, which is called the Lumley triangle. Fluctuations of turbulence can be described by analyzing the anisotropy tensor *a_ij_* in Equations (4) and (5).
(4)$$a_{ij}=\overset{-}{u_{i}^{\prime }u_{j}^{\prime }}/2k-\delta_{ij}/3.$$
where $u_{i}^{\prime }$ is the instant velocity fluctuation along the *i* direction, $k=\overset{-}{u_{n}^{\prime }u_{n}^{\prime }}/2$ is the turbulent kinetic energy, *δ_ij_* is the Kronecker delta function, and the tensor *a_ij_* has three scalar invariants:(5)$$I=a_{ij}=0,II=a_{ij}a_{ji},III=a_{ij}a_{jk}a_{ki}.$$

Through calculation, the distributions of the AIMs at different gap-to-diameter ratios when *Re* = 7.02 × 10^4^ and *r** = 0.8 are shown by the red, blue and green spots in Figure 8. Under all three working conditions, the scalar III is approximately 0, so the turbulence is in the state of the plane-strain limit [29].

### 3.4. Skin Friction Coefficient

Figure 9 displays the curves of the mean value of the rotor’s skin friction coefficient *C_f_** (including the upper and lower faces and the cylindrical face of the rotor) with *Re* at three gap-to-diameter ratios, where (b) is the partial enlarged view of the area in the purple box in (a). The skin friction coefficient increases with *Re* under all three working conditions. When *Re* < 10^4^, the *C_f_** decreases as the gap-to-diameter ratio *G* increases, as shown in Figure 9b, and as *Re* continues to increase, the *C_f_** at *G* = 0.04 gradually becomes the maximum, while the *C_f_** at *G* = 0.027 becomes the minimum value when *Re* is greater than 9.6 × 10^4^. The main reason may be that at a low *Re*, the friction coefficients of the end faces and the cylindrical face are in the same order of magnitude, and that at a greater gap-to-diameter ratio they are in a smaller velocity gradient, making the *C_f_** decrease as the gap-to-diameter ratio increases. When *Re* increases, the stress in the radial slit changes more significantly at a greater gap-to-diameter ratio, causing the friction coefficient of the cylindrical face to significantly increase and the velocity of such increase to be greater than that of the friction coefficient of the end faces, making the *C_f_** reach its maximum when the gap-to-diameter ratio *G* = 0.04. Moreover, as *Re* increases, the velocity gradient changes more significantly at a smaller gap-to-diameter ratio, so *C_f_** at *G* = 0.013 gradually exceeds that at *G* = 0.027. For the liquid-floating rotor micro gyroscope in this work, the higher the rotating speed, the more significant the gyroscopic effect and the greater the frictional resistance, resulting in noticeable power consumption and heating. Under the three working conditions in this study, a gap-to-diameter ratio *G* of 0.027, i.e., an axial slit of 200 μm, is the optimal choice.

### 3.5. Flow Vortex Structure Distribution in the Radial Gap

An increase in *Re* may change the flow distribution of the flow field in the axial gap, and also affect the vortex structure in the radial gap. The model is divided along the *r*-*z* plane to observe the distribution of the Taylor vortex on the section, as shown in Figure 10. At a low *Re*, the Taylor vortex in the radial gap is integrated with the circulating flow in the axial gap, and as *Re* increases, they are gradually separated from each other, because the viscosity force decreases, and the laminar flow becomes unstable. When *Re* increases to 7.02 × 10^4^, four independent areas of Taylor turbulence appear in the axial gap, which are of an approximately symmetric distribution.

The cloud image of the intensity distribution of the spiral vortices on different faces in the *θ*-*z* plane is shown in Figure 11. The spiral vortex near the cylindrical face of the rotor is significantly more intense than that near the lateral wall of the cavity. At a low *Re*, most of the spiral vortices are near the axial slit of the rotor. As *Re* increases, the range of the spiral vortices gradually expands and covers the center of the flow field. When *Re* increases to 7.02 × 10^4^, the spiral vortices fluctuate approximately periodically, and gradually fill the entire flow field.

## 4. Conclusions

In this paper, an RSC microscale flow field simulation model of liquid-floating rotor micro gyroscopes is built to calculate the flow field distribution at different values of *Re* and gap-to-diameter ratios with the RSM. The mean flow, turbulence statistics, anisotropic distribution of turbulence and the change laws of the flow vortex structure and frictional resistance coefficient in the radial gap have been attained.

(1)Along the z-axis, the circumferential velocity gradually develops from a linear distribution to a non-linear distribution with separated boundary layers, i.e., from a torsional Couette flow to a Batchelor flow, as *Re* increases. The greater the *Re* is, the more the velocity gradient in the middle part tends to 0. The local *Re* (*Re_l_*) and the gap-to-diameter ratio (*G*) affect the boundary layers in different ways. The former mainly affects the velocity distribution near the stationary boundary, and an increase in it causes an increase in the velocity gradient. The latter mainly affects the velocity distribution near the rotational wall, and an increase in it causes a decrease in the velocity gradient. The radial velocity is in an S-shaped distribution under all three working conditions. Due to the centrifugal force, the fluid near the rotational wall flows outward along the radius, while that near the stationary wall flows inward along the radius to supplement the former. Both *Re* and the gap-to-diameter ratio affect the extremum of the radial velocity and its position along the z-axis. Compared with the circumferential velocity and the radial velocity, the axial velocity is quite small, nearly 0 in the axial slit.(2)The Reynolds stresses are mainly in an Σ-shaped and lateral U-shaped distribution at different values of *Re* and gap-to-diameter ratios. Most of the areas of turbulence are near the stationary boundary layer and the rotational boundary layer. When *Re* > 10^4^, *Re* has little effect on the distribution of the Reynolds stresses along the z-axis. As the gap-to-diameter ratio increases, the maximum of the Reynolds normal stress increases, and the Reynolds shear stress has no obvious law. According to the analysis results of the anisotropy invariants of the turbulence under different working conditions, the turbulence in this work is in the state of the plane-strain limit.(3)The skin friction coefficient increases as *Re* increases. When *Re* < 10^4^, the greater the gap-to-diameter ratio, the smaller the friction resistance coefficient. If *Re* > 10^4^, it increases the most slowly with the gap-to-diameter ratio *G* = 0.027. When *Re* increases to 9.6 × 10^4^, the skin friction coefficient is smaller with *G* = 0.027 than that with *G* = 0.04 and 0.013.(4)As *Re* increases, the flow vortex structure in the radial gap gradually separates. When *Re* increases from 1.75 × 10^3^ to 7.02 × 10^4^, the vortex structure in the radial gap gradually separates from the circulating flow in the axial gap, and eventually forms four independent vortices in the radial gap. An increase in *Re* causes the range of spiral vortices to gradually expand from the cylindrical face of the rotor to the lateral wall of the cavity. When *Re* ≥ 7.02 × 10^4^, periodic fluctuations appear in spiral vortices in the radial gap.

## Figures and Tables

**Figure 1 micromachines-14-00793-f001:**
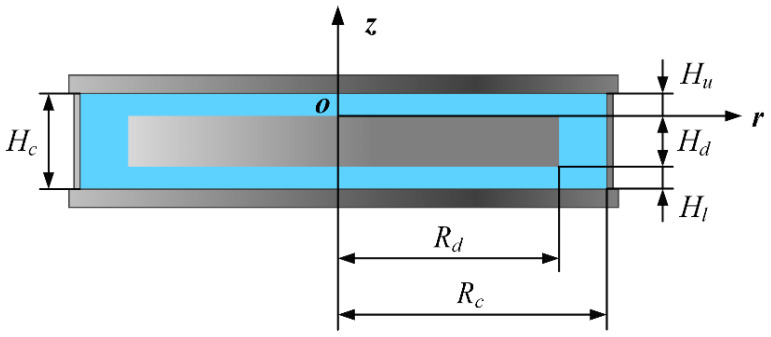
Schematic diagram of the gemetry in the *r*-*z* plane.

**Figure 2 micromachines-14-00793-f002:**
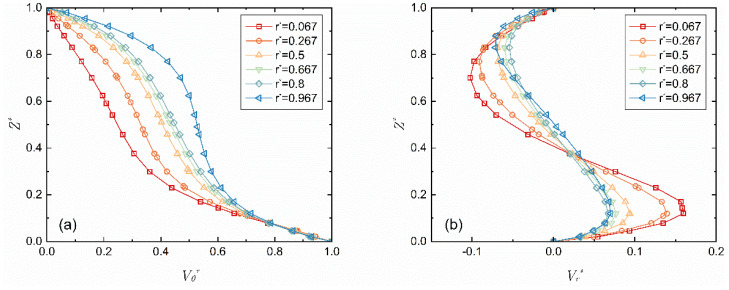
Distribution curve of mean velocity along the z-axis when *Re* = 7.02 × 10^4^ (12,000 rpm) and *G* = 0.027. (**a**) Circumferential velocity; (**b**) Radial velocity.

**Figure 3 micromachines-14-00793-f003:**
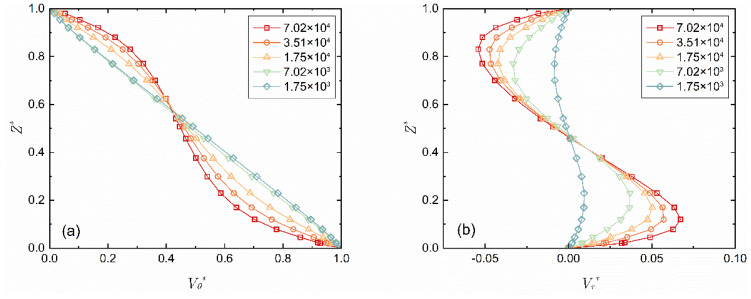
Distribution curve of the velocity component along the z-axis when *r** = 0.8 and *G* = 0.027. (**a**) Circumferential velocity; (**b**) Radial velocity.

**Figure 4 micromachines-14-00793-f004:**
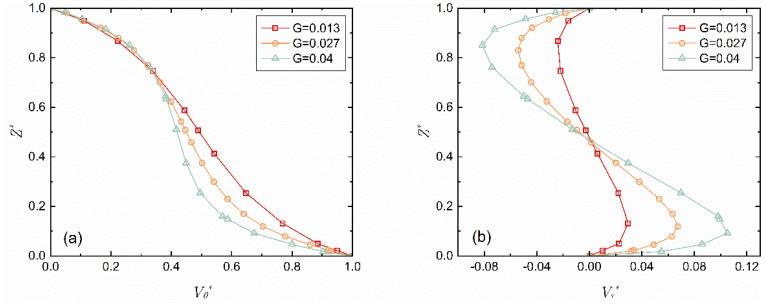
Distribution curve of the velocity component along the axis-z when *r** = 0.8 and *Re* = 7.02 × 10^4^. (**a**) Circumferential velocity; (**b**) Radial velocity.

**Figure 5 micromachines-14-00793-f005:**
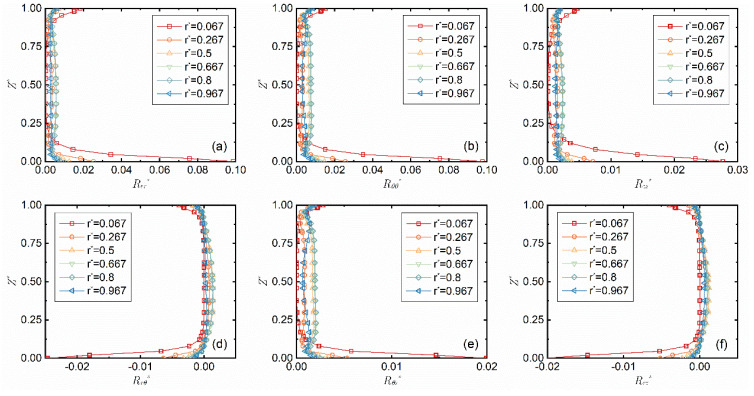
Change of the six components of the Reynolds stress tensor (**a**–**f**) with the radial position *r** when *Re* = 7.02 × 10^4^ and *G* = 0.027.

**Figure 6 micromachines-14-00793-f006:**
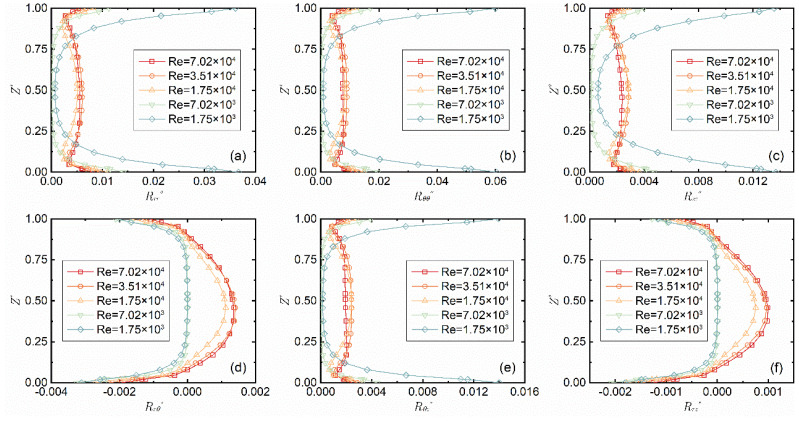
Change of the six components of the Reynolds stress tensor (**a**–**f**) with *Re* when *G* = 0.02 and *r** = 0.8.

**Figure 7 micromachines-14-00793-f007:**
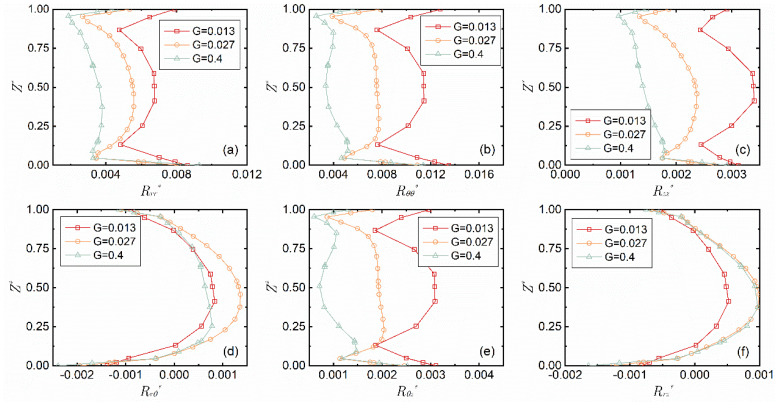
Change of the six components of the Reynolds stress tensor (**a**–**f**) with the gap-to-diameter ratio *G* when *Re* = 7.02 × 10^4^ and *r** = 0.8.

**Figure 8 micromachines-14-00793-f008:**
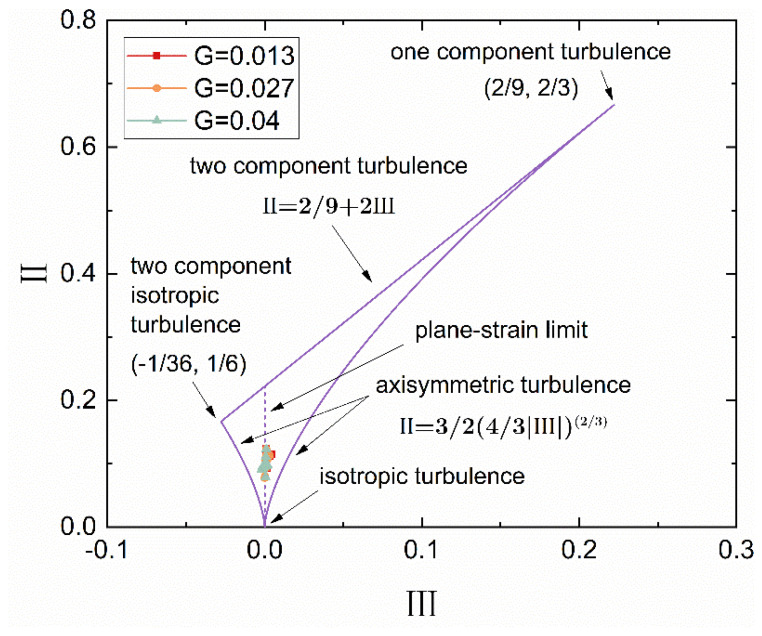
The anisotropy invariant map.

**Figure 9 micromachines-14-00793-f009:**
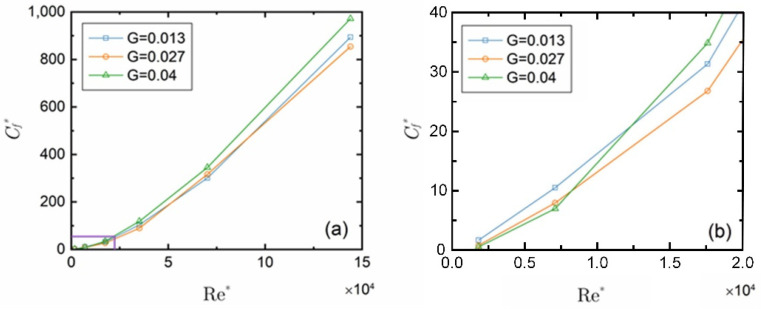
Change of the skin friction coefficient of the rotor with *Re*: (**a**) Overall view; (**b**) Partial enlarged view.

**Figure 10 micromachines-14-00793-f010:**
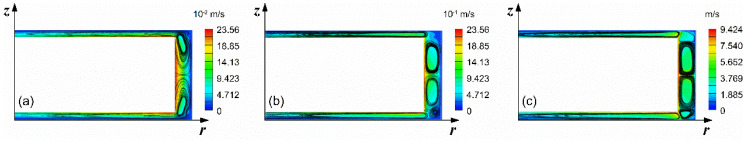
Distribution of the Taylor turbulence at different *Re.* (**a**) *Re* = 1.75 × 10^3^; (**b**) *Re* = 1.75 × 10^4^; (**c**) *Re* = 7.02 × 10^4^.

**Figure 11 micromachines-14-00793-f011:**
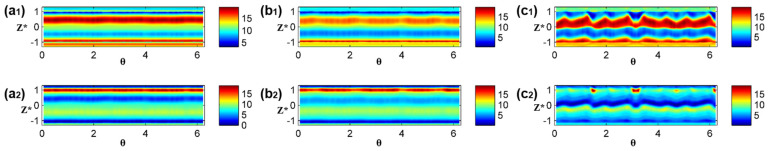
Cloud image of intensity distribution of spiral vortices. (**a_1_**,**b_1_**,**c_1_**) are 10 μm to the lateral wall of the rotor; (**a_2_**,**b_2_**,**c_2_**) are 10 μm to the lateral wall of the cavity; (**a**) *Re* = 1.75 × 10^3^; (**b**) *Re* = 1.75 × 10^4^; (**c**) *Re* = 7.02 × 10^4^.

**Table 1 micromachines-14-00793-t001:** Model parameters.

*R_d_* (mm)	*H_d_* (mm)	*R_c_* (mm)
7.5	2	8

**Table 2 micromachines-14-00793-t002:** Values of relaxation factor for different parameters.

Parameter	Pressure	Body Force	Momentum	Turbulent Kinetic Energy
Relaxation Factor	0.3	0.5	0.4	0.4

## Data Availability

The data that support the findings of this study are available from the corresponding author upon reasonable request.

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
