# Peer review of "Simulations of the Rotor-Stator-Cavity Flow in Liquid-Floating Rotor Micro Gyroscope"

_micromachines, 2023, doi:10.3390/mi14040793_

Round 1

Reviewer 1 Report

In this paper, an RSC microscale flow field simulation is established to calculate the flow field distribution at different Re numbers and gap-to-diameter ratios with the Reynolds stress model (RSM). The mean flow, turbulence statistics and frictional resistance coefficient in axial slits and the distribution law of the vortex structure in radial slits have been attained.

I have the following questions and suggestions:

1. Has grid independence verification been done? What are the criteria for evaluation?

2. The paper mentions k- ω model and RSM model. Have the two models been calculated and compared?

3. Write the specific meaning of r in the formula Rel in the paper.

4. What is the G value of the fluid domain model in Part 3.5 of the paper? When the G value changes, is the Taylor vortex separation consistent?

Author Response

Reviewer 1: 

  1. Has grid independence verification been done? What are the criteria for evaluation?

Re: Thanks for your professional question. 

The grid independence verification has been done to ensure the efficiency and accuracy of the solution calculation. The radial scale of the simulated flow field is much larger than the axial scale, so the model is divided into upper and lower end caps, upper and lower ring domains, and cylindrical areas. Each area is meshed separately. The grids near the rotor wall and the cavity shell wall are refined. The grid orthogonality quality was used for evaluation. The closer the value is to 1, the better the grid quality. The overall mesh quality distribution is shown in Fig.1, as following. The grid quality distribution is concentrated at 0.8, indicating that the grid quality is better.

Fig.1. Distribution map of mesh quality

  1. The paper mentions k- ω model and RSM model. Have the two models been calculated and compared?

Re: Thanks for your professional question. 

The applicable scenarios of different turbulence models have beensorted in the reference book as Tab.1.[1] The k-ω model is better for the simulation of wall boundary layer and low Reynolds number flow, and RSM is suitable for complex 3D flow simulation of strong swirling flow, such as rotating flow and curved pipe flow. Therefore, this paper selects the RSM model directly.

Tab.1. The applicable scenarios of different turbulence models

Model

Applicable Scenario

SAM

The amount of calculation is small, and it is mostly used for flow simulation in the aviation field

Standard k-ε

The amount of calculation is medium and it is suitable for high Reynolds numbers, and the results are poor when simulating rotation and flow

RNG k-ε

It can simulate separation flow, jet flow, secondary flow, etc., but can not accurately simulate strong swirling flow

Realizable k-ε

Except for strong rotating flow, other flow models can be simulated well

Standard k-ω

It is good for wall boundary layer and low Reynolds number flows

RSM

It is suitable for complex 3D flow simulation of strong swirling flow, such as rotating flow, curved pipe flow, etc. Compared with other models, it has obvious advantages

  1. Write the specific meaning of r in the formula Rel in the paper.

Re: Thanks for your professional suggestion.

r in the formula  represents radial distance from the origin and has been supplemented in Section 2 Model parameters and numerical method.

  1. What is the G value of the fluid domain model in Part 3.5 of the paper? When the G value changes, is the Taylor vortex separation consistent?

Re: Thanks for your professional question. 

G value is 0.027 in Part 3.5 of the paper. Under the three values of G (0.013, 0.027, 0.04) studied in this paper, the Taylor vortex separation was consistent.

Reference:

[1] Pengfei Li, Minyi Xu, FeiFei Wang. Proficient in CFD Engineering Simulation and Case Practice: FLUENT GAMBIT ICEM CFD Tecplot. Post & Telecom Press, Bei Jing, China, 2011.

Reviewer 2 Report

1. First, I have a serious question about the laminar flow. How did the authors indicate the laminar flows from the RSM model developed for the full developed turbulent flows?  I mean the sentences like

the laminar flow on the line 151,

the laminar flow becomes unstable. When Re increases on the line 244

2. Second nothing is said on the code and numerical setup used in the paper.

3. Third, the word separation has a definite meaning in fluid mechanics. I suppose the authors mean something becomes different in sentences like

...the rotational boundary layer gradually separates from the stationary boundary layer 

... separation gradually occurs. According to the above results, there is no core region with   on the line 117

4. Usage of some words is not typical. Samples:

Vortexes--> vortices,  Reynolds meanà Reynolds averaged, Ekaman boundary layerà Ekman

5. Typos:

unchanged.II   what is II on line 131

6. Graphical presentation

Fig. 1. Schematic diagram of the flow field in the r-z plane. This is not the flow field this is Sketch of the geometry under study in the r-z plane

Fig.11  It would be good to show designations along vertical and horizontal axis

Author Response

  1. First, I have a serious question about the laminar flow. How did the authors indicate the laminar flows from the RSM model developed for the full developed turbulent flows?  I mean the sentences like

the laminar flow on the line 151,

the laminar flow becomes unstable. When Re increases on the line 244?

Re: Thanks for your professional question.

Theoretically, the Reynolds stress is closely related to the fluctuating velocity, and it can be written as following:

(1)

the unclosed turbulent diffusion must be modeled to close the equation when solving. It has been simplified in ANSYS Fluent to use a scalar turbulent diffusivity as following:

                            (2)

In laminar flow with low Reynolds number, the fluctuating velocity is very small, almost zero, so the non-closed term calculated by the model also tends to zero. Therefore, when the Reynolds number is low, the RSM fails and does not affect the calculation of the laminar flow. Therefore, the RSM can be used for simulation at low Reynolds numbers.

  1. Second nothing is said on the code and numerical setup used in the paper.

Re: Thanks for your professional suggestion. 

In this paper, ANSYS Fluent software is used for simulation.

First,the 3D model was built. The radial scale of the simulated flow field is much larger than the axial scale, so the model is divided into upper and lower end caps, upper and lower ring domains, and cylindrical areas.

Second, the grid independence verification had been done to ensure the efficiency and accuracy of the solution calculation. Each area was meshed separately with structured grid. The grids near the rotor wall and the cavity shell wall were refined. The grid orthogonality quality was used for evaluation. The closer the value is to 1, the better the grid quality. The grid quality distribution is concentrated at 0.8, indicating that the grid quality is better.

Fig.1. Distribution map of mesh quality

Third,convenience conditions and initial conditions were set. The upper and lower end surfaces and side walls of the rotor were set as the rotation boundary and the rotation speed is set to 300rpm, 1200rpm, 3000rpm, 6000rpm, 12000rpm, etc. All the walls of the cavity shell were set as the static boundary. All walls were set as no-slip boundary conditions. The liquid in the model is set to be water at 20°C.

Fourth, the parameters of solver were set. Pressure, momentum, turbulent kinetic energy, and turbulent diffusion rate adopted the standard, the second-order upwind, the first-order upwind, and the first-order upwind, respectively. After discretization, the SIMPLE algorithm was used to solve the pressure coupling equation. The equations solved by Fluent are non-linear, and only by controlling the variable value in each iteration can the divergence caused by large difference be effectively avoided. It is regulated by the relaxation factor. After testing, when the relaxation factor of each calculation parameter was set to Tab. 1, the calculation was easy to converge.

Tab.1 Values of relaxation factor for different parameters

Parameter

Pressure

Body Force

Momentum

Turbulent kinetic energy

relaxation factor

0.3

0.5

0.4

0.4

Last, iterative calculations were performed. The convergence standard of the residual value of each variable is set to 10-6. In addition, the velocity of a certain point in the model is selected as a reference. When it reached stability, the calculation result was considered to have converged.

These above description have been added in Section 2 Model parameters and numerical method.

  1. Third, the word separation has a definite meaning in fluid mechanics. I suppose the authors mean something becomes different in sentences like

...the rotational boundary layer gradually separates from the stationary boundary layer

... separation gradually occurs. According to the above results, there is no core region with   on the line 117.

Re: Thanks for your professional suggestion.

Separation in this paper means that boundary layer changes as the flow regime changes. As the Reynolds number increases, the distribution of the velocity field in the z-axis direction changes and is no longer a linear distribution. The flow regime changes from torsional Couette flow with merged boundary layer to Batchelor flow with unmerged boundary layer. In Batchelor flow, there is a Bödewadt boundary layer near the rotating wall and an Ekman boundary layer near the stationary wall.

  1. Usage of some words is not typical. Samples:

Vortexes--> vortices,  Reynolds mean-->Reynolds averaged, Ekaman boundary layer--> Ekman

Re: I am sorry about that. These atypical words have been revised.

  1. 5. Typos: unchanged.II   what is II on line 131

Re: I am sorry about that. This is a typo and II has been deleted.

  1. Graphical presentation

Fig. 1. Schematic diagram of the flow field in the r-z plane. This is not the flow field this is Sketch of the geometry under study in the r-z plane

Fig.11  It would be good to show designations along vertical and horizontal axis

Re: Thanks for your professional suggestion. The questions of this two pictures has been revised.

Reviewer 3 Report

Dear authors. 

Enclosed are my comments. 

Regards. 

Author Response

  1. 1. Strongly improve the literature review by including some contextualization about wall turbulence and generally fluid modeling in engineering. In particular, why RSM is better? Did the authors try other turbulence models? Here is an example of several way to model turbulence in near-wall conditions. I suggest to have a look at them and frame them the introduction:

https://doi.org/10.1016/j.cja.2014.12.007

https://doi.org/10.1016/j.compfluid.2022.105710

https://doi.org/10.1016/j.ijheatfluidflow.2022.109071

https://doi.org/10.1017/S0022112010003113

Re: Thanks for your professional suggestion. 

We have carefully read the recommended references, added the reasons for choosing RSM, and framed them into Section 1 instruction.

2.Section 2 must be dramatically improved by describing in detail the model characteristics. In particular,

- the software used is not provided

- the mesh is not described

- no mesh sensitivity is given.

How can I be sure of the results obtained?

Is there any comparison available with experimental data?

Other simulations?

Other models?

Re: Thanks for your professional suggestion.

In this paper, ANSYS Fluent software was used for simulation.

First,the 3D model was built. The radial scale of the simulated flow field is much larger than the axial scale, so the model is divided into upper and lower end caps, upper and lower ring domains, and cylindrical areas.

Second, the grid independence verification had been done to ensure the efficiency and accuracy of the solution calculation. Each area was meshed separately with structured grid. The grids near the rotor wall and the cavity shell wall were refined. The grid orthogonality quality was used for evaluation. The closer the value is to 1, the better the grid quality. The grid quality distribution is concentrated at 0.8, indicating that the grid quality is better.

Fig.1. Distribution map of mesh quality

Third,convenience conditions and initial conditions were set. The upper and lower end surfaces and side walls of the rotor were set as the rotation boundary and the rotation speed is set to 300rpm, 1200rpm, 3000rpm, 6000rpm, 12000rpm, etc. All the walls of the cavity shell were set as the static boundary. All walls were set as no-slip boundary conditions. The liquid in the model were set to be water at 20°C.

Fourth, the parameters of solver were set. Pressure, momentum, turbulent kinetic energy, and turbulent diffusion rate adopted the standard, the second-order upwind, the first-order upwind, and the first-order upwind, respectively. After discretization, the SIMPLE algorithm was used to solve the pressure coupling equation. The equations solved by Fluent are non-linear, and only by controlling the variable value in each iteration can the divergence caused by large difference be effectively avoided. It is regulated by the relaxation factor. After testing, when the relaxation factor of each calculation parameter was set to Tab. 1, the calculation was easy to converge.

Tab.1. Values of relaxation factor for different parameters

Parameter

Pressure

Body Force

Momentum

Turbulent kinetic energy

relaxation factor

0.3

0.5

0.4

0.4

Last, iterative calculations were performed. The convergence standard of the residual value of each variable is set to 10-6. In addition, the velocity of a certain point in the model is selected as a reference. When it reaches stability, the calculation result is considered to have converged.

These above description have been added in Section 2 Model parameters and numerical method.

The closed flow field in micro-scale and high-speed rotating device are being built now. The simulation and experimental results will be compared after the experimental conditions meeting the requirements.

According to the reference book[1], We sorted out the applicable scenarios of different turbulence models, as Tab.2. RSM is more suitable for RSCs flow, and we choose it directly.

Tab.2. The applicable scenarios of different turbulence models

Model

Applicable Scenario

SAM

The amount of calculation is small, and it is mostly used for flow simulation in the aviation field

Standard k-ε

The amount of calculation is medium and it is suitable for high Reynolds numbers, and the results are poor when simulating rotation and flow

RNG k-ε

It can simulate separation flow, jet flow, secondary flow, etc., but can not accurately simulate strong swirling flow

Realizable k-ε

Except for strong rotating flow, other flow models can be simulated well

Standard k-ω

It is good for wall boundary layer and low Reynolds number flows

RSM

It is suitable for complex 3D flow simulation of strong swirling flow, such as rotating flow, curved pipe flow, etc. Compared with other models, it has obvious advantages

Reference:

[1] Pengfei Li, Minyi Xu, FeiFei Wang. Proficient in CFD Engineering Simulation and Case Practice: FLUENT GAMBIT ICEM CFD Tecplot. Post & Telecom Press, Bei Jing, China, 2011.

Round 2

Reviewer 2 Report

I am still not satisfied with the idea to calculate laminar- turbulent flows using models developed for full developed turbulent flows because many of closing formulas are obtained under condition of the full developed turbulence. For transitional flows the transitional models should be used. Therefore, the approach utilized by the authors can be used as a very rough estimation.

see video:

https://www.youtube.com/watch?v=5htknS9uVEk

Reviewer 3 Report

Dear authors. I am ok with this version of the paper.